# Multimode Decomposition and Wavelet Threshold Denoising of Mold Level Based on Mutual Information Entropy

**DOI:** 10.3390/e21020202

**Published:** 2019-02-21

**Authors:** Zhufeng Lei, Wenbin Su, Qiao Hu

**Affiliations:** School of Mechanical Engineering, Xi’an Jiaotong University, 28 West Xianning Road, Xi’an 710049, Shaanxi, China

**Keywords:** variational mode decomposition, wavelet threshold, empirical mode decomposition, denoising, mutual information entropy

## Abstract

The continuous casting process is a continuous, complex phase transition process. The noise components of the continuous casting process are complex, the model is difficult to establish, and it is difficult to separate the noise and clear signals effectively. Owing to these demerits, a hybrid algorithm combining Variational Mode Decomposition (VMD) and Wavelet Threshold denoising (WTD) is proposed, which involves multiscale resolution and adaptive features. First of all, the original signal is decomposed into several Intrinsic Mode Functions (IMFs) by Empirical Mode Decomposition (EMD), and the model parameter *K* of the VMD is obtained by analyzing the EMD results. Then, the original signal is decomposed by VMD based on the number of IMFs *K*, and the Mutual Information Entropy (MIE) between IMFs is calculated to identify the noise dominant component and the information dominant component. Next, the noise dominant component is denoised by WTD. Finally, the denoised noise dominant component and all information dominant components are reconstructed to obtain the denoised signal. In this paper, a comprehensive comparative analysis of EMD, Ensemble Empirical Mode Decomposition (EEMD), Complementary Empirical Mode Decomposition (CEEMD), EMD-WTD, Empirical Wavelet Transform (EWT), WTD, VMD, and VMD-WTD is carried out, and the denoising performance of the various methods is evaluated from four perspectives. The experimental results show that the hybrid algorithm proposed in this paper has a better denoising effect than traditional methods and can effectively separate noise and clear signals. The proposed denoising algorithm is shown to be able to effectively recognize different cast speeds.

## 1. Introduction

In the modern steel industry, efficient continuous casting technologies, including key continuous casting equipment and new processes, have become the core technology. The continuous casting process is a complex and continuous phase change process. There are many factors that affect the quality of slabs. The research on the core technologies in the high-quality steel continuous casting process is mainly focused on the precision of the mold, the sector, and the composition control of molten steel [1].

The mold is referred to as the heart of the continuous casting machine. The mold level control is the basis of stable production operation to avoid break out and steel overflow at the mold top. The fluctuation of the mold level is disturbed by various factors and has negative characteristics of nonlinearity, variation over time, and uncertainty. These include time-varying and nonlinear disturbances in production operations such as abrupt changes in casting speed and the slide gate owing to wear and clogging.

Precise mold level monitoring is regarded as the key to improving continuous casting production quality as shown in Figure 1 [2,3,4]. It is an important source of reference data for casting speed control, segment roll gap control, mold cooling water control, and stopper rod opening control. If the mold level fluctuates too much, the following will occur. First, it will cause impurities on the surface of the mold. Surface defects and internal defects of the slab are generated which affect the surface and internal quality of the slab. Second, it will affect the casting speed, affecting productivity and the production rhythm. Eventually, it will cause the slab and the continuous casting machine to stick together, damage the tundish slide, and even cause downtime. Accurate prediction of the mold level occupies an important position in the continuous casting production process. This paper proposes an advanced mold level signal denoising method to prepare accurate data input for future mold level prediction, realize the purpose of predictive control, and greatly reduce the occurrence of accidents affecting quality and safety in the continuous casting production process.

In order to maintain the stability of the mold level, scholars have conducted much research on mold level control. The main methods include proportion integration differentiation (PID) control, fuzzy control, and adaptive control. Michel Dussud et al. developed a fuzzy controller based on expert knowledge for process control when the crystallizer level is disturbed [5]. T. Hesketh et al. applied an adaptive controller to the control of the mold level of a continuous casting mold, providing a new method of control [6]. Robin De Keyser et al. introduced the application of a model predictive control to mold level control, providing a new approach for mold level control [7]. RMC De Keyser introduced a new method based on automatic tuning and predictive control to improve the control of the mold level [8]. F. Kong et al. conducted a simulation and study on the performance of different adaptive predictive control methods and compared the merits and demerits of the different methods [9]. Regardless of the control methods, the first thing that must be done is collecting and analyzing the data of the continuous casting. However, the continuous casting production process is a strong coupling and nonlinear process with a large number of interference signals, resulting in a decrease in the accuracy of the control method.

In the continuous casting production process, accurate modeling of process parameters usually requires noise-free data. However, due to the fact that the process of data collection and transmission is subject to various noises, accurate data cannot be obtained. As a result, an accurate system model is difficult to establish.

In recent years, many signal processing methods have been applied to the signal denoising process. Many scholars have done much research on signal denoising. JS Smith presented the results of applying local mean decomposition (LMD) to a scalp electroencephalogram (EEG) visual perception dataset. The analysis suggests that there is a statistically significant difference between the theta phase concentrations of perception and no perception EEG data [10]. M.G. Frei et al. presented intrinsic time-scale decomposition (ITD) for efficient and precise time–frequency–energy (TFE) analysis of signals [11]. In 1998, Huang et al. proposed empirical mode decomposition (EMD) decomposition [12]. EMD is an adaptive decomposition method without a prior matrix [13]. Since the introduction of EMD by Huang, it has been widely used in biomedicine [14,15], speech recognition [16], system modeling [17,18,19], and process control [20,21]. It can decompose a signal into several Intrinsic Mode Functions (IMFs) in order from high frequency to low frequency. Noise is generally considered to be mainly concentrated on high frequency IMFs, so denoising can be achieved by removing high frequency IMFs. EMD is widely used for signal denoising. Manas et al. [22] used EMD and ASMF (Adaptive Switching Mean Filter) for ECG signal joint denoising. W Chen et al. proposed improved EMD algorithms such as Ensemble Empirical Mode Decomposition (EEMD) and Empirical Wavelet Transform (EWT) to denoise seismic data and demonstrated good performance [23,24,25]. D.M. Klionskiy et al. discussed pattern discovery in signals via EMD and the EMD technique relative to signal denoising; they concluded that EMD is an efficient tool for signal denoising in the case of homoscedastic and heteroscedastic noise [26,27]. Butusov D et al. proposed a new filtering algorithm based on the cascade of driven chaotic oscillators, and the algorithm showed the best performance and reliability compared with traditional denoising and filtering approaches [28]. Wavelet technology [29] has multiresolution analysis characteristics and good time–frequency locality, and clear signals and noise can be separated according to different characteristics of wavelet coefficients. Therefore, the wavelet threshold method is considered to be a fast, reasonable, and effective method for denoising. Wavelet-based technology has been widely applied to signal denoising. Based on Wavelet Threshold Denoising (WTD) technology, Mingkun Su et al. [30] dealt with the multipath interference problem of high precision positioning of global navigation satellite systems. Zhu. Q [31] presented a joint denoising method for coal seam hydraulic fracturing micro seismic signals based on multi-threshold wavelet packets and EMD. In recent years, a new signal processing method, the Variational Mode Decomposition (VMD) technique, has enriched the signal denoising method. In 2014, Konstantin Dragomiretskiy et al. [32] proposed the variational mode decomposition method. VMD is a completely nonrecursive variational mode decomposition model. It finds the center frequency and bandwidth of each decomposition component by iteratively searching for the optimal solution of the variational model and adaptively splits the frequency domain of the signal, effectively separating the components. Qiyang Xiao et al. [33] proposed a phase denoising method based on digital speckle interferometry with improved VMD. The research results show that this method can effectively filter out noise interference, and the peak signal-to-noise ratio is higher than those of other noise reduction methods. Siwei Yu et al. [34] introduced VMD into seismic random noise attenuation data, overcoming the low resolution of empirical mode decomposition; the numerical results show that the method based on VMD performs much better than the method based on EMD, especially in terms of preserving the tail.

Recent studies have shown that although there are many methods in the field of signal denoising processing, none of them are applicable to all data. The wavelet-based signal denoising method is widely used, but the wavelet denoising method is limited by the selection of the wavelet basis, which affects the generalization ability of wavelet denoising. Although the EMD-based method is widely used for the adaptability of its decomposition, the EMD method has serious mode aliasing and boundary effects, which seriously affect the signal decomposition. Particularly, in the process of noise reduction processing, the EMD method often directly removes high frequency components, resulting in the loss of valid information. The signal denoising technique based on the VMD method has been widely used in recent years. Compared with the EMD method, VMD effectively avoids mode aliasing and boundary effects and can adaptively split the frequency domain of the signal and effectively separate the components. The VMD-WTD hybrid algorithm combines the advantages of VMD and WTD. This method avoids modal aliasing and boundary effects and also has multiresolution and good time–frequency locality.

This paper focuses on the application of the hybrid algorithm to signal denoising and applies it to the denoising of a mold level to ensure the quality of mold level data. First, the mold level signal is decomposed into several Intrinsic Mode Functions (IMFs) by EMD, and by analysis of the Mutual Information Entropy (MIE) between IMFs, the mode number *K* of the VMD is determined. Then, the mold level signal is decomposed into *K* IMFs by VMD, and analysis of the MIE between IMFs is used to identify the noise dominant components and the information dominant components. Next, in order to avoid the loss of effective information, the noise dominant components are denoised by the WTD, and the effective information is properly retained. Finally, all information components and the denoised noise dominant components are reconstructed to obtain the denoised signal. The remainder of this paper is organized as follows. The second section introduces the VMD algorithm and MIE. The third section introduces the VMD-WTD algorithm. Section 4 compares the performance of eight algorithms; the last section is the conclusion.

## 2. Basic Algorithm Research

### 2.1. Variational Mode Decomposition (VMD)

VMD is a new signal decomposition estimation method based on classical Wiener filtering, Hilbert transform, and mixing. This method determines the frequency center and bandwidth of each decomposition component by iteratively searching for the optimal solution of the variational model, and it can also decompose the signal into components with sparsity characteristics adaptively [32]. The VMD algorithm redefines an amplitude–frequency modulation function as an intrinsic mode function; the expression is as follows(1)$$u_{k}(t)=A_{k}(t)\cos(\phi_{k}(t))$$
where phase *ϕ_k_*(*t*) is a nondecreasing function, *A_k_*(*t*) is the instantaneous amplitude of *u_k_*(*t*), and *A_k_*(*t*) ≥ 0. *ω_k_*(*t*) = *ϕ’_k_*(*t*), which is the instantaneous frequency of *u_k_*(*t*).

In the interval range of [*t* − δ, *t* + δ], *u_k_*(*t*) can be regarded as a harmonic signal with amplitude *A_k_*(*t*) and frequency ω*_k_*(*t*), and $\delta =2\pi /{\phi^{\prime }}_{k}(t)$.

The variational constraint model is as follows(2)$$\begin{array}{l} \underset{\left\{u_{k}\},\{\omega_{k}\right\}}{\min}\{{\sum_{k}\left\|\partial_{t}[\left(\delta (t)+\frac{j}{\pi t}\right)\times u_{k}(t)]e^{-j\omega kt}\right\|}_{2}^{2}\}\\ s.t.\sum_{k}u_{k}=f \end{array}$$
where $\left\{u_{k}\right\}:=\left\{u_{1},u_{2},\ldots u_{K}\right\}$ is the number of IFMs; $\left\{\omega_{k}\right\}:=\left\{\omega_{1},\omega_{2},\ldots ,\omega_{K}\right\}$ is the frequency center of each IMF; and $\sum_{k}:=\sum {}_{k=1}^{K}$ is the sum of all modes.

We introduce the Lagrange function as(3)$$L\left(\left\{u_{k}\right\},\left\{\omega_{k}\right\},\lambda \right)=\alpha {\sum_{k}\left\|\partial_{t}[\left(\delta (t)+\frac{j}{\pi t}\right)\times u_{k}(t)]e^{-j\omega kt}\right\|}_{2}^{2}+{\left\|f(t)-\sum_{k}u_{k}(t)\right\|}_{2}^{2}+\langle \lambda (t),f(t)-\sum_{k}u_{k}(t)\rangle$$
where *α* is the penalty factor and *λ* is the Lagrange multiplier. ${\left\|f(t)-\sum_{k}u_{k}(t)\right\|}_{2}^{2}$ is the second penalty.

The problem of solving the original minimum value can be transformed into the saddle point of the extended Lagrange expression by the alternating direction method, which is the optimal solution of the above formula.(4)$$u_{k}^{n+1}=\arg_{uk}\min L(\left\{u_{i<k}^{n+1}\right\},\left\{u_{i\geq k}^{n+1}\right\},\left\{\omega_{i}^{n}\right\},\lambda^{n})$$
(5)$$\omega_{k}^{n+1}=\arg_{\omega k}\min L(\left\{u_{i}^{n+1}\right\},\left\{\omega_{i<k}^{n+1}\right\},\left\{\omega_{i\geq k}^{n}\right\},\lambda^{n})$$
(6)$$\lambda^{n+1}=\lambda^{n}+\tau \left(x-\sum_{k}u_{k}^{n+1}\right)$$
where ${{\sum_{k}\left\|u_{k}^{n+1}-u_{k}^{n}\right\|}_{2}^{2}/\left\|u_{k}^{n}\right\|}_{2}^{2}<\varepsilon$ is convergence condition. n is the number of iterations.

Therefore, the original signal can be decomposed into *K* IMFs.

The calculation process of the VMD Algorithm 1 is listed as follows.**Algorithm 1** The calculation process of the VMDStep 1: Initialize {uk1}、{ωk1}、λ1 and *n* to zero;Step 2: *n* = *n* + 1, execute the entire loop;Step 3: Execute the loop *k* = *k* + 1 until *k* = *K*, update uk: ukn+1=argminLuk({ui<kn+1},{ui≥kn},{uin},λn);Step 4: Execute the loop *k* = *k* + 1, until *k* = *K*, update ωk:ωkn+1=argminLωk({ωi<kn+1},{ωi≥kn},{ωin},λn);Step 5: Use λn+1=λn+τ(f(t)−∑kuk(t)) to update *λ*;Step 6: Given the discrimination condition ε > 0, if the iteration stop condition is satisfied, all the cycles are stopped and the result is output; *K* IMFs are obtained.

When obtaining the IMFs, the VMD method gets rid of the iterative screening process used by the EMD method. Instead, it processes the signal through the variational model principle and solves the adaptive decomposition of the signal by solving the optimal solution of the constrained variational model. In order to take the bandwidth of the modal function into account, the following provisions are required. (1) For each intrinsic mode function *u_k_*, the Hilbert transform is applied to calculate the correlation analysis signal in order to obtain a single sideband spectrum. (2) For each modality, the center frequency is estimated separately by mixed exponential tuning, and the modal frequency spectrum is transferred to the baseband. (3) The bandwidth is estimated by demodulating the H Gaussian smoothness of the signal.

### 2.2. Mutual Information Entropy (MIE)

After VMD decomposition, the mold level signal *s*(*t*) is as follows(7)$$s(t)=\sum_{i=1}^{n}S_{IMF_{i}}(t)+r_{n}(t)$$
where $S_{IMF_{i}}(t)$ represents the IMF, whose frequency is ranked from high to low; *r_n_*(*t*) represents the residual, representing the average trend of the signal.(8)s(t)=H(t)+L(t)

$H(t)=\sum_{i=1}^{k-1}S_{IMF_{i}}(t)$ represents the IMF combination of high frequency IMFs.

$L(t)=\sum_{i=1}^{n}S_{IMF_{i}}(t)+r_{n}(t)$ represents the IMF and residual combination of low frequency IMFs.

MIE is used to measure the statistical dependence between two random variables. The expression is as follows(9)$$I(X,Y)=\sum_{i=1}^{r}\sum_{j=1}^{s}p(x_{i},y_{i})lb\frac{p(x_{i},y_{i})}{p(x_{i})p(y_{i})}$$
where *p*(*x_i_*, *y_i_*) is the joint probability distribution; *p*(*x_i_*) and *p*(*y_i_*) are the edge probability distributions; *X* and *Y* represent different IMFs; and *r* and *s* represent the number of symbols of *X* and *Y*.

We assume that the high frequency component is noise interference and the low frequency component is a valid signal. The MIE between the IMFs can be used to identify the boundary between the high frequency and low frequency IMFs. The IMF characteristics obtained by the VMD show that the dependence of the high frequency noise on each IMF is gradually reduced, and the dependence of the low frequency effective signal on each IMF component is gradually enhanced. Therefore, it can be assumed that the high frequency component and the low frequency component are partially statistically independent of each other. It is known from the characteristics of MIE that the MIE between two independent random variables should be equal to 0 [35]. Therefore, when calculating the MIE between each adjacent IMF, a local minimum value occurs, and only the first local minimum value is searched to obtain a boundary between the high frequency component and the low frequency component. Thus, the search objective function is as follows(10)$$k=first\{\underset{1\leq i\leq n-1}{\min}\left[I(S_{IMF_{i}},S_{IMF_{i+1}})\right]\}$$
where *K* is the serial number of the high frequency and low frequency components decomposed into IMFs.

## 3. Denoising Algorithm Using VMD-WTD

A single modal decomposition method has a certain effect on the signal-to-noise separation of noisy signals, but there are also various problems that may cause loss of valid information and result in incomplete noise removal. In this paper, a hybrid denoising algorithm which can separate the clear signal and noise and preserve the valid information as much as possible is proposed to further denoise the noise-dominated modal component generated by single mode decomposition. Besides this, it can also improve the denoising effect and performance indicators of the modal decomposition method. The VMD-WTD denoising flowchart is shown in Figure 2.

Since the mode number of the original signal decomposition in the VMD algorithm is an a priori knowledge estimation, there is a certain randomness which may cause errors in the mode decomposition. Based on the characteristics of EMD adaptive decomposition, without decomposing the mode number, the original signal is decomposed, the IMFs are observed and obtained, the MIE between IMFs is analyzed, and the effective mode number *K* is determined.

The variational mode decomposition (VMD)–wavelet threshold algorithm is listed as Algorithm 2. EMD: empirical mode decomposition; VMD: variational mode decomposition; IMF: intrinsic mode functions; WTD: wavelet threshold denoising; MIE: mutual information entropy.**Algorithm 2** The variational mode decomposition (VMD)–wavelet thresholdStep 1. Decompose the original signal with EMD to obtain several IMFs.Step 2. Analyze the MIE between IMFs and determine the effective mode number *K*.Step 3. Decompose the original signal by VMD based on the mode number *K* determined by the EMD.Step 4. Analyze the MIE between IMFs and find the boundary line between high frequency IMFs and low frequency IMFs.Step 5. Denoise high frequency IMFs with WTD, retaining low frequency IMFs and retaining effective information as much as possible.Step 6. Perform VMD reconstruction on all low frequency IMFs and denoise high frequency IMFs to obtain a denoised signal.

## 4. Numerical Experiments

### 4.1. Analog Signal Test

In order to verify the effect of the method described in this paper, the simulation data were tested. The original data included four impact signals. The equation is as follows*y*_1_ = cos(2*π* × *f*_1_*t*) + 0.3cos(2*π* × *f*_2_*t*) + 0.02cos(2*π* × *f*_3_*t*)*n* = 0.5*randn*(*t*)*y* = *y*_1_ + *n*
where *f*_1_ = 4, *f*_2_ = 45, and *f*_3_ = 150; *y*_1_ is a clear signal with different amplitudes of different frequencies; and *y* is a noisy signal. The noisy signal and clear signal are shown in Figure 3.

Results of EMD is shown in Figure 4. As shown in Table 1, the MIE of IMF (2–3) is 4.02, which is the first local minimum of the MIE between IMFs by EMD. We determined that IMF3 is the boundary between high frequency IMFs and low frequency IMFs, with IMF1 and IMF2 as one mode, the other IMFs each acting as a mode separately, and *K* = 6. Then, the original data were decomposed by VMD based on *K* = 6.

Results of VMD is shown in Figure 5. As shown in Table 2, the MIE of IMF (3–4) is 3.70. This is the first local minimum of MIE between IMFs by VMD, and IMF4 is considered to be the boundary line between high frequency IMFs and low frequency IMFs; WTD was performed on the first four IMFs.

Results of WTD is shown in Figure 6. After WTD, the center frequencies of the first four IMFs were significantly reduced, and the amplitudes were also significantly reduced.

As shown in Figure 7, the noise frequency of the noisy signal is 77 Hz and the noise frequency of the denoised signal is 77 Hz which has no energy. This method has a good denoising effect and effectively retains the effective information of the original signal.

### 4.2. Mold Level Data Source

In order to show the applicability and superiority of the hybrid denoising method proposed, this paper adopts the actual process parameters collected from the continuous casting machine (HBIS Group Hansteel company, Handan, China) developed by the China Heavy Machinery Research Institute for noise reduction. The parameter is the data of the mold level in the continuous casting process. The control of the mold level is very important in the continuous casting system and is of great significance for the quality and casting safety. Due to many uncertain disturbance factors in the mold level control process, the disturbance may change constantly, and most of the disturbances are nonlinear and nonstationary. Clear representation of the mold level data is of great significance for improving the process parameters of continuous casting production. The realization of the denoising method proposed in this paper has important practical significance.

### 4.3. Denoising the Mold Level Signal Using VMD-WTD 

The EMD method was used to decompose the mold level signal into nine IMFs and one residual, as shown in Figure 8.

As shown in Table 3 and Figure 8, the MIE of IMF (2–3) is 0.7567, which is the first local minimum MIE between IMFs by EMD, so IMF3 is the boundary line between high frequency IMFs and low frequency IMFs. The high frequency IMFs are seen as a mode, while the other IMFs are seen as different modes separately, and we determined that *K* = 9. The mold level signal was decomposed using VMD based on *K* = 9.

As shown in Table 4 and Figure 9, the MIE of IMF (4–5) is 1.4114, which is the first local minimum MIE between IMFs by VMD, so IMF 5 is the boundary line between the high frequency IMFs and the low frequency IMFs; we performed WTD denoising on the first five IMFs.

As shown in Figure 9, by selecting *K* = 9 as the mode component number for VMD decomposition we can clearly separate the original signals and avoid modal aliasing. It can be seen from the spectrum diagram that the IMF3–IMF7 frequency bandwidth is relatively long and the noise is serious, so WTD was performed on these IMFs. 

As shown in Figure 10, the first five IMFs’ center frequencies were significantly reduced and the amplitudes were also significantly reduced.

It can be seen from Figure 10 that the IMFs after WTD have a more pronounced center frequency and a much narrower frequency band. IMF1–IMF5 and the other IMFs after WTD were reconstructed using VMD to obtain the denoised signal.

### 4.4. The Discussion of Results

The performance of eight methods was verified using two statistical indicators in this paper, and the method that is most suitable for mold level was selected.

Root-Mean-Square Error (*RMSE*):(11)$$R=\frac{Cov(P,A)}{\sqrt{Var[P]\cdot Var[A]}}.$$

Signal–Noise Ratio (*SNR*):(12)$$SNR=10\log_{10}(\frac{P_{i}}{A_{i}}).$$

In Equations (11) and (12), *P_i_* and *A_i_* are the *i*-th denoised value and actual value, respectively, and *n* is the total number of signal points.

As shown in Table 5 and in Figure 11 and Figure 12, in the three basic algorithms, the denoising effect of VMD is better than that of the others, which is due to the good frequency division function of the VMD. The frequency division of VMD is equivalent to a low-pass filter, which avoids the nonadaptation of the WTD algorithm for wavelet basis selection and avoids the modal aliasing and boundary effects caused by EMD adaptive decomposition. The EMD adaptive decomposition provides a better reference of the number of modal components for VMD decomposition, which further enhances the robustness of the VMD algorithm to noise.

A single algorithm forcibly removing high frequency components cannot avoid losing valid information. The VMD-WTD algorithm is an improvement on the VMD algorithm and avoids the loss of valid information caused by forced removal of high frequency components. In comparison with the two algorithms in the hybrid, the denoising effect of VMD-WTD is better. The VMD-WTD denoising algorithm proposed in this paper shows improved denoising performance compared with the other seven algorithms analyzed, showing strong generalization ability and robustness.

## 5. Application of the Mold Level Denoised Signal

First, 40 groups of mold level signals under three different casting speeds were selected. After the signals were processed by the algorithm proposed in this paper, the maximum energy IMF in each set of data was calculated, and the center frequency of the maximum energy IMF was selected. As a feature, the center frequency distribution of the data before and after the denoising process is shown in Figure 13, and the proposed algorithm can effectively distinguish different casting speeds in the continuous casting process in order to monitor changes in the casting speed.

## 6. Conclusions

This paper proposed a novel denoising algorithm which combines multiple mode decomposition algorithms, MIE, and the WTD algorithm, which is an adaptive denoising algorithm. The algorithm uses multiple mode decomposition methods, takes MIE as the threshold for identifying noise, and uses the WTD algorithm as the main denoising algorithm. As shown through the simulation and comparison of simulation data and measured signals, the denoising algorithm has the following advantages.(1)This is the first multimode decomposition denoising algorithm to be proposed.(2)It is a new denoising algorithm using multimode decomposition and WTD to be applied to mold level control.(3)In comparison with other algorithms, the proposed algorithm is a better denoising algorithm with higher SNR and lower RMSE.(4)By using the denoising algorithm and feature extraction method proposed in this paper, the center frequency information is determined. Compared with the feature extraction information without denoising, the experimental results show that the proposed denoising algorithm can effectively improve the recognition of the casting speed.

## Figures and Tables

**Figure 1 entropy-21-00202-f001:**
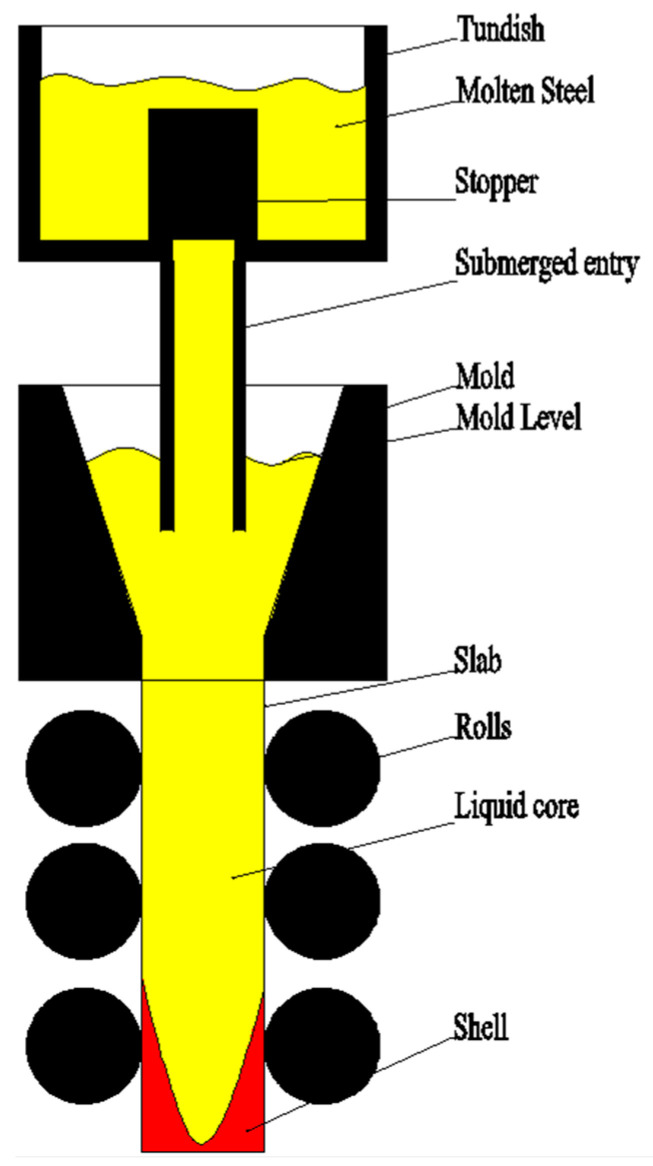
Mold level model.

**Figure 2 entropy-21-00202-f002:**
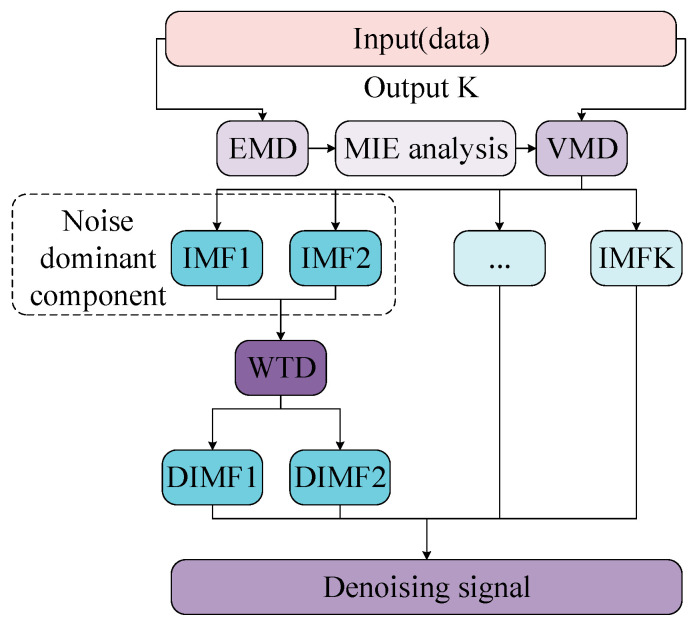
VMD-WTD denoising flowchart. EMD: empirical mode decomposition; VMD: variational mode decomposition; IMF: intrinsic mode functions; WTD: wavelet threshold denoising; MIE: mutual information entropy.

**Figure 3 entropy-21-00202-f003:**
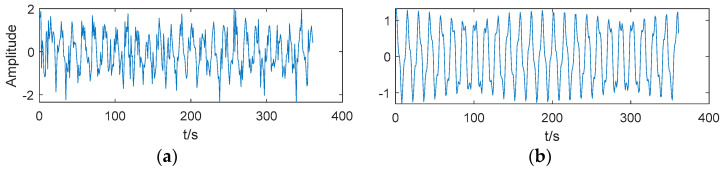
(**a**) Noisy signal. (**b**) Clear signal.

**Figure 4 entropy-21-00202-f004:**
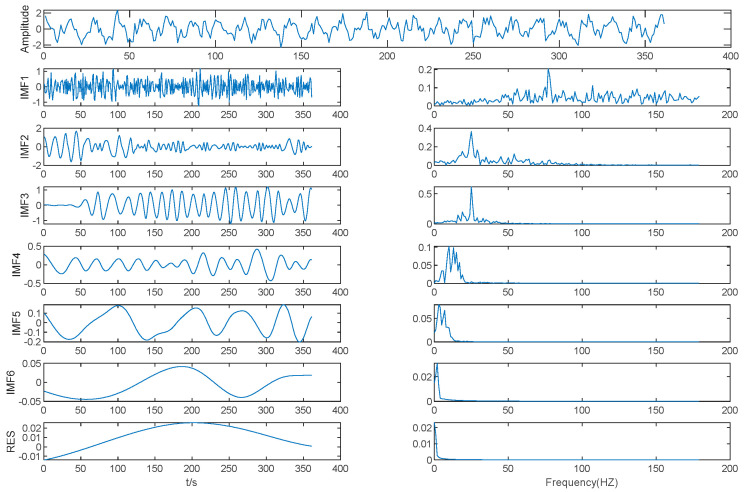
Results of EMD.

**Figure 5 entropy-21-00202-f005:**
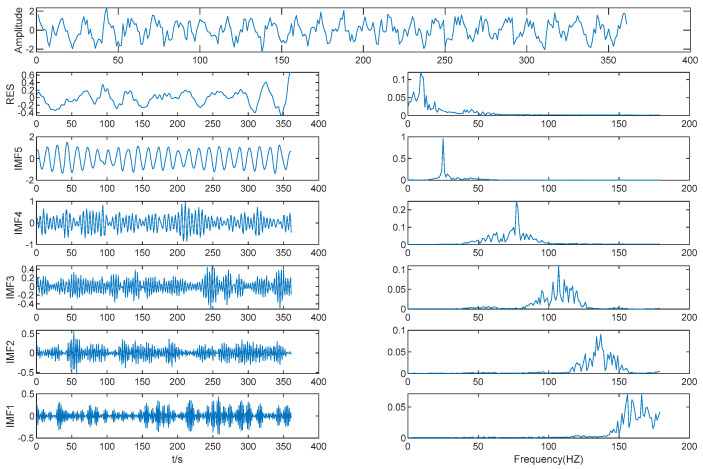
Results of VMD.

**Figure 6 entropy-21-00202-f006:**
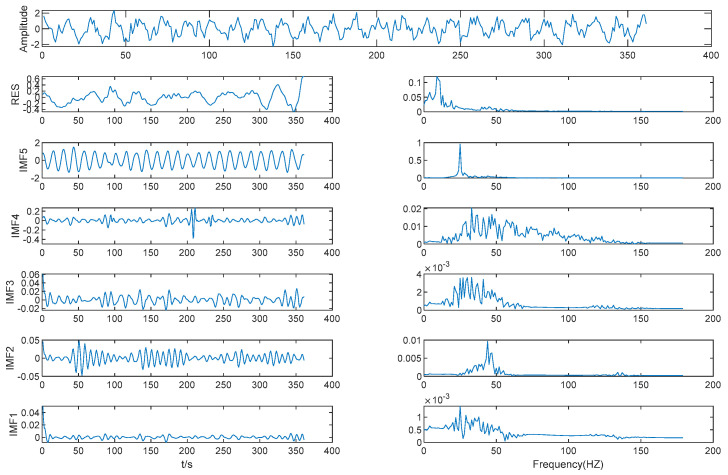
Results of WTD.

**Figure 7 entropy-21-00202-f007:**
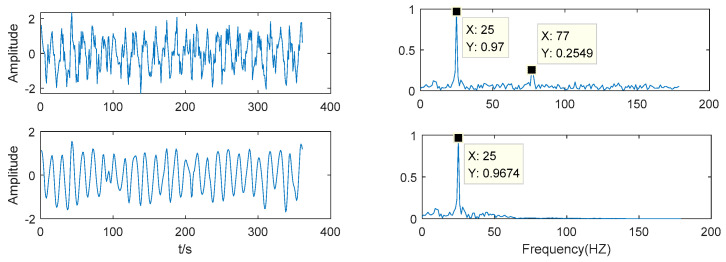
Comparison after VMD-WTD decomposition.

**Figure 8 entropy-21-00202-f008:**
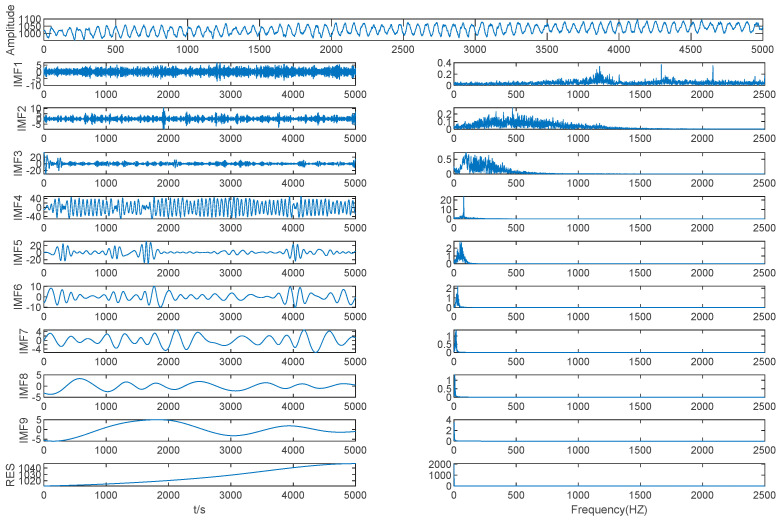
Mold level decomposition results of EMD.

**Figure 9 entropy-21-00202-f009:**
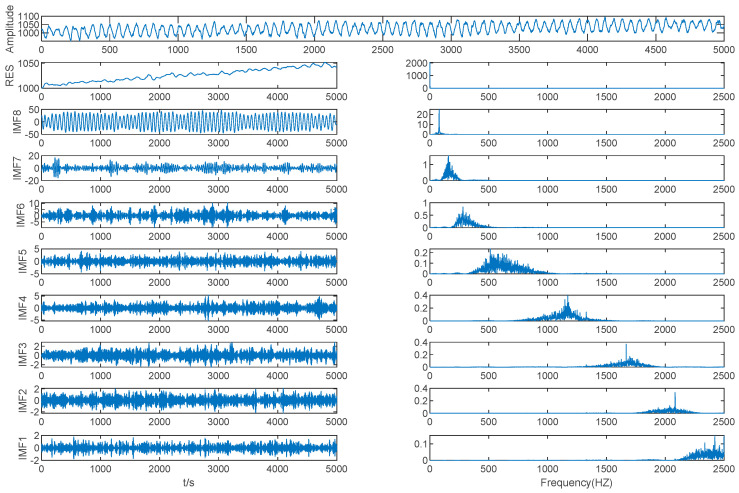
Mold level decomposition result by VMD.

**Figure 10 entropy-21-00202-f010:**
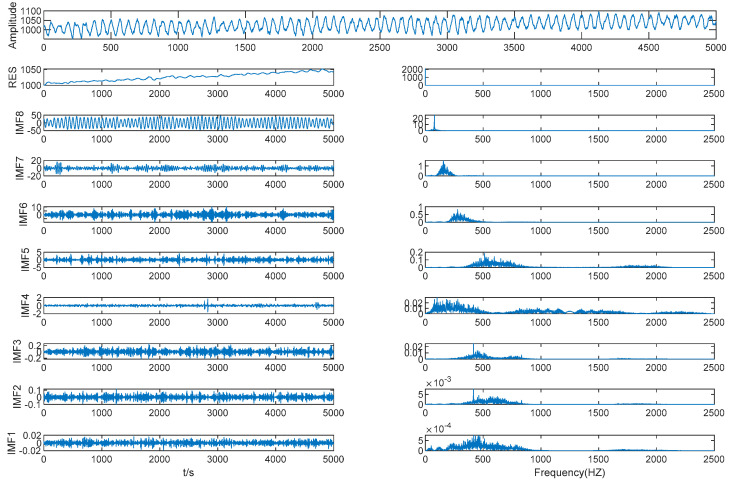
WTD result of IMF1–IMF5.

**Figure 11 entropy-21-00202-f011:**
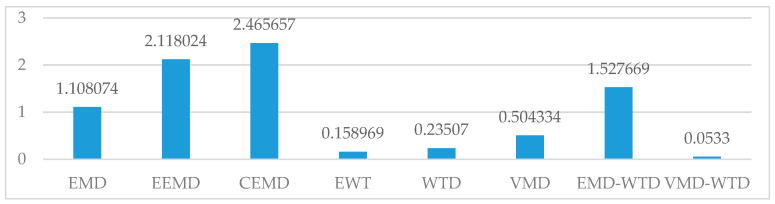
Root-Mean-Square Error (RMSE) indicator for denoising results of multiple algorithms. EEMD: ensemble empirical mode decomposition; CEEMD: complete ensemble empirical mode decomposition; EWT: empirical wavelet transform.

**Figure 12 entropy-21-00202-f012:**
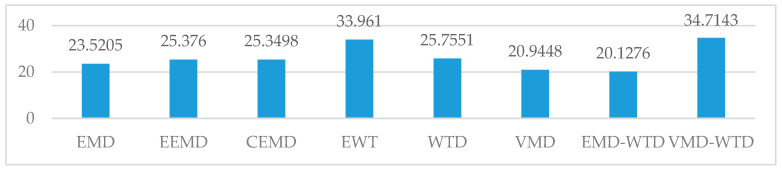
SNR indicator for denoising results of multiple algorithms. EEMD: ensemble empirical mode decomposition; CEEMD: complete ensemble empirical mode decomposition; EWT: empirical wavelet transform.

**Figure 13 entropy-21-00202-f013:**
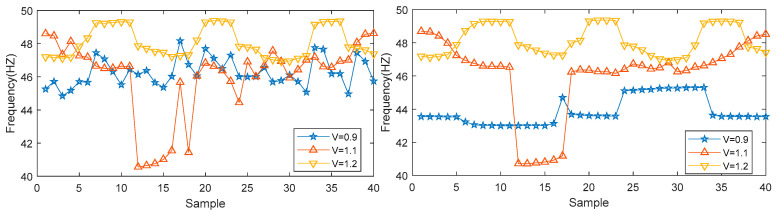
Distribution of maximum energy IMF center frequency.

**Table 1 entropy-21-00202-t001:** MIE between IMFs by EMD. res is residual.

IMF (1–2)	IMF (2–3)	IMF (3–4)	IMF (4–5)	IMF (5–6)	IMF (6–res)
4.07	4.02	4.11	4.01	4.03	4.25

**Table 2 entropy-21-00202-t002:** MIE between IMFs by VMD.

IMF (1–2)	IMF (2–3)	IMF (3–4)	IMF (4–5)	IMF (5–res)
3.77	3.83	3.70	4.02	3.89

**Table 3 entropy-21-00202-t003:** MIE between IMFs by EMD.

IMF (1–2)	IMF (2–3)	IMF (3–4)	IMF (4–5)	IMF (5–6)	IMF (6–7)	IMF (7–8)	IMF (8–9)	IMF (9–res)
1.1859	0.7567	1.2681	1.6978	1.7153	2.2602	2.5477	3.1388	4.1474

**Table 4 entropy-21-00202-t004:** MIE between IMFs by VMD.

IMF (1–2)	IMF (2–3)	IMF (3–4)	IMF (4–5)	IMF (5–6)	IMF (6–7)	IMF (7–8)	IMF (8–res)
1.6254	1.6488	1.4727	1.4114	1.4858	1.4476	1.8755	2.2944

**Table 5 entropy-21-00202-t005:** Denoising results.

	EMD	EEMD	CEEMD	EWT	WTD	VMD	EMD-WTD	VMD-WTD
RMSE	1.108074	2.118024	2.465657	0.158969	0.23507	0.504334	1.527669	0.0533
RNS	23.5205	25.376	25.3498	33.9610	25.7551	20.9448	20.1276	34.7143

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
