# Peer review of "Multimode Decomposition and Wavelet Threshold Denoising of Mold Level Based on Mutual Information Entropy"

_entropy, 2019, doi:10.3390/e21020202_

Round 1

Reviewer 1 Report

Dear authors, please find my comments below.

First of all, I recommend to significantly improve the style and grammar of the manuscript.

Second, I do not understand, why the manuscript submitted for publication in MDPI Entropy has Sensors 2019, 19, x; doi: FOR PEER REVIEW in subscriptions. These little things make the paper look sloppy. 

Figure 1 should be clarified - the text is poorly readable.

In my opinion, the introduction is excessively inflated and can be reduced. The novelty of your approach and the contribution of your paper must be emphasized.

Equation 2 is not clear for me. If you mean the fist local minima, then please correct the description, which says "

How did you obtain K=6 implementing EMD for 6 modes?

Why do you analyze MIE? It does not impact the K calculation.

Why the IMF border #3 is К=6? Shouldn't it be К=3? If I got it right, K is the real number of modes.

I heavily doubt that a multi-mode decomposition denoising algorithm is proposed for the first time in this paper, please clarify this or refer to:

1. Klionskiy D., Kupriyanov M., Kaplun D. Signal denoising based on empirical mode decomposition //Journal of Vibroengineering. Volume 19, Issue 7, 1 November 2017, Pages 5560-5570. DOI: 10.21595/jve.2017.19239. 

2. D. M. Klionskiy, D. I. Kaplun, V. V. Geppener. Empirical Mode Decomposition for Signal Preprocessing and Classification of Intrinsic Mode Functions // Pattern Recognition and Image Analysis. January 2018, Volume 28, Issue 1, pp 122–132. https://doi.org/10.1134/S1054661818010091.

3. Butusov D., Karimov T., Voznesenskiy A., Kaplun D., Andreev V., Ostrovskii V. Filtering Techniques for Chaotic Signal Processing // Electronics, Vol. 7.12. №450 (DOI: 10.3390/electronics7120450, WOS:000455067800109)

"In comparison with other algorithms, the proposed algorithm is a better denoising algorithm with higher signal-to-noise ratio and lower RMSE". 

Please consider more algorithms in this comparison. 

Taking all of this into account, my decision is major revisions.

Author Response

Thanks to the reviewers for their hard work and careful guidance.

I have already responded to the reviewer’s comments as follows.

Point 1: I recommend to significantly improve the style and grammar of the manuscript.   

Response 1: Reference reviewer's opinion, I have asked a native English speaker to edit the file, and I have modified the error that the editor pointed out in the paper.

Point 2: I do not understand, why the manuscript submitted for publication in MDPI Entropy has Sensors 2019, 19, x; doi: FOR PEER REVIEW in subscriptions. These little things make the paper look sloppy.

Response 2: I am sorry for my carelessness, I have made relevant changes.

Point 3: Figure 1 should be clarified - the text is poorly readable. In my opinion, the introduction is excessively inflated and can be reduced. The novelty of your approach and the contribution of your paper must be emphasized.

Response 3: Reference reviewer's opinion, I have revised related content and figure. The revised content is as follows: Line 50: “If the mold level fluctuates too much, the following will occur: First, it will cause impurities on the surface of the mold. Surface defects and internal defects of the slab are generated which affect the surface and internal quality of the slab. Second, it will affect the casting speed, affecting productivity and the production rhythm. Eventually it will cause the slab and the continuous casting machine to stick together, damage the tundish slide, and even cause downtime. Accurate prediction of the mold level occupies an important position in the continuous casting production process. This paper proposes an advanced mold level signal denoising method to prepare accurate data input for future mold level prediction, realize the purpose of predictive control, and greatly reduce the occurrence of accidents affecting quality and safety in the continuous casting production process.”.

Figure 1

Point 4: Equation 2 is not clear for me. If you mean the fist local minima, then please correct the description, which says "

How did you obtain K=6 implementing EMD for 6 modes?

Why do you analyze MIE? It does not impact the K calculation.

Why the IMF border #3 is К=6? Shouldn't it be К=3? If I got it right, K is the real number of modes.

Response 4: EMD is adaptive decomposition. The number of IMFs decomposed by EMD is not affected by humans. The VMD process is not adaptive decomposition. The number of IMFs decomposed by VMD is determined by K value. The determination of K by human affects the effect of VMD decomposition. So, using the adaptive decomposition of EMD, MIE analysis of IMFs after EMD can effectively determine the value of K. The VMD results of K=3, K=4, K=5, K=6 and K=7 are given below.

As shown in above figures, when K6, there is mode aliasing in IMFs. When K=6 (obtained by the method proposed in this paper), the centre frequency of each IMF is obvious, and no pattern aliasing occurs. When K6, there is mode aliasing in IMFs too. The result shows that MIE is used to analyse the result of EMD to determining K is correct.

Point 5: I heavily doubt that a multi-mode decomposition denoising algorithm is proposed for the first time in this paper, please clarify this or refer to:

1. Klionskiy D., Kupriyanov M., Kaplun D. Signal denoising based on empirical mode decomposition //Journal of Vibroengineering. Volume 19, Issue 7, 1 November 2017, Pages 5560-5570. DOI: 10.21595/jve.2017.19239.

2. D. M. Klionskiy, D. I. Kaplun, V. V. Geppener. Empirical Mode Decomposition for Signal Preprocessing and Classification of Intrinsic Mode Functions // Pattern Recognition and Image Analysis. January 2018, Volume 28, Issue 1, pp 122–132. https://doi.org/10.1134/S1054661818010091.

3. Butusov D., Karimov T., Voznesenskiy A., Kaplun D., Andreev V., Ostrovskii V. Filtering Techniques for Chaotic Signal Processing // Electronics, Vol. 7.12. №450 (DOI: 10.3390/electronics7120450, WOS:000455067800109)?

Response 5: Reference reviewer's opinion, I have revised related content and references. The revised content is as follows: Line 92: “D. M. Klionskiy et al. discusses pattern discovery in signals via EMD and the EMD technique relative to signal denoising. They conclude that EMD is an efficient tool for signal denoising in the case of homoscedastic and heteroscedastic noise [26,27]. Butusov D et al. proposed a new filtering algorithm based on the cascade of driven chaotic oscillators, and the algorithm showed the best performance and reliability than traditional denoising and filtering approaches [28]”

References [26].

References [27].

References [28].

Point 6: "In comparison with other algorithms, the proposed algorithm is a better denoising algorithm with higher signal-to-noise ratio and lower RMSE".

Please consider more algorithms in this comparison.

Response 6: Reference reviewer's opinionI have added related content. The revised content is as follows: Line 308:

"Figure 11. RMSE indicator for denoising results of multiple algorithms"

"Figure 12. SNR indicator for denoising results of multiple algorithms"

Table 6 Denoise results

EMD

EEMD

CEEMD

EWT

WTD

VMD

EMD-WTD

VMD-WTD

RMSE

1.108074

2.118024

2.465657

0.158969

0.23507

0.504334

1.527669

0.0533

RNS

23.5205

25.376

25.3498

33.9610

25.7551

20.9448

20.1276

34.7143

Reviewer 2 Report

The manuscript applies the algorithm in signal processing to the precision mold liquid level monitoring, which has certain application innovation, and the predicted results are also in good agreement with the actual. My comments are as follows:
(1) The paper conducts a comparative study based on EMD. Does the author compare the results with EMD's improved algorithms, such as EEMD, CEEMD, EWT, etc.? Can the author give reasons for comparative analysis based on EMD? Or give a data analysis algorithm that is best suited to precision mold level data. I suggest the authors consider the methods being similar to EMD in the following references,and cite them in the introduction to better introduce the development in related fields.

[1]Automatic noise attenuation based on clustering and empirical wavelet transform. Journal of Applied Geophysics, 2018, 159: 649-665. 

[2]Ground roll attenuation using improved complete ensemble empirical mode decomposition. Journal of Seismic Exploration, 2016, 25(5): 485-495.             

[3]Random noise reduction using a hybrid method based on ensemble empirical mode decomposition. Journal of Seismic Exploration, 2017, 26(3): 227-249. 

[4]J.S. Smith.The local mean decomposition and its application to EEG perception data. Journal of the Royal Society Interface, 2(5) (2005) 443-454.

[5]M.G. Frei. Intrinsic time-scale decomposition: time-frequency-energy analysis and real-time filtering of non-stationary signals. Proceedings Mathematical Physical & Engineering Sciences, 463 (2078) (2007) 321 -342.

(2) Is the analysis data given in the article a signal in the time domain? Is EMD suitable for non-time domain data analysis? I think the noise level of the data in Figure 3 is too low. The author should discuss the new method of data noise suppression in the background of strong noise, which may be more innovative. If the data in the paper contains too little noise, then there will be some conceptual error between data decomposition and denoising.

Author Response

Thanks to the reviewers for their hard work and careful guidance.

I have already responded to the reviewer’s comments as follows.

Point 1: The paper conducts a comparative study based on EMD. Does the author compare the results with EMD's improved algorithms, such as EEMD, CEEMD, EWT, etc.? Can the author give reasons for comparative analysis based on EMD? Or give a data analysis algorithm that is best suited to precision mold level data. I suggest the authors consider the methods being similar to EMD in the following references, and cite them in the introduction to better introduce the development in related fields.

[1]Automatic noise attenuation based on clustering and empirical wavelet transform. Journal of Applied Geophysics, 2018, 159: 649-665.

[2]Ground roll attenuation using improved complete ensemble empirical mode decomposition. Journal of Seismic Exploration, 2016, 25(5): 485-495.            

[3]Random noise reduction using a hybrid method based on ensemble empirical modedecomposition. Journal of Seismic Exploration, 2017, 26(3): 227-249.

[4]J.S. Smith.The local mean decomposition and its application to EEG perception data. Journal of the Royal Society Interface, 2(5) (2005) 443-454.

[5]M.G. Frei. Intrinsic time-scale decomposition: time-frequency-energy analysis and real-time filtering of non-stationary signals. Proceedings Mathematical Physical & Engineering Sciences, 463 (2078) (2007) 321 -342..   

Response 1: Reference reviewer's opinionI have added related content and references. The revised content is as follows: Line 78: “JS Smith presented the results of applying local mean decomposition (LMD) to a set of scalp electroencephalogram (EEG) visual perception data. The analysis suggests that there is a statistically significant difference between the theta phase concentrations of the perception and no perception EEG data [10]. M.G. Frei et al. presented intrinsic time-scale decomposition (ITD) to efficient and precise time–frequency–energy (TFE) analysis of signals [11]”.

Line 90: “W Chen et al. proposed improved EMD algorithms, such as Ensemble Empirical Mode Decomposition (EEMD) and Empirical Wavelet Transform (EWT) to denoised the seismic data, demonstrated good performance [23-25]”

References [10].

References [11]. 

References [23].

References [24].

References [25].

In addition, I have added a comparison of several methods.

"Figure 11. RMSE indicator for denoising results of multiple algorithms"

"Figure 12. SNR indicator for denoising results of multiple algorithms"

Table 6 Denoise results

EMD

EEMD

CEEMD

EWT

WTD

VMD

EMD-WTD

VMD-WTD

RMSE

1.108074

2.118024

2.465657

0.158969

0.23507

0.504334

1.527669

0.0533

RNS

23.5205

25.376

25.3498

33.9610

25.7551

20.9448

20.1276

34.7143

Point 2: Is the analysis data given in the article a signal in the time domain? Is EMD suitable for non-time domain data analysis? I think the noise level of the data in Figure 3 is too low. The author should discuss the new method of data noise suppression in the background of strong noise, which may be more innovative. If the data in the paper contains too little noise, then there will be some conceptual error between data decomposition and denoising.

Response 2: Thanks to the reviewer's suggestion, the data analysed in this paper is a time series. Whether EMD is suitable for non-time series, I have not studied it, I dare not just draw conclusions. For the noise problem in Figure 3, because this is simulation signal, due to itself the clear signal has a small amplitude. If the noise energy is too large, the clear signal cannot be resolved. In addition, the reviewer's analysis in a strong noise environment is a very interesting part, and I will continue to study in the next step.

Round 2

Reviewer 1 Report

Thank you for revising the paper. All my comments were properly addressed, thus I will recommend this manuscript to be accepted. However, some proofreading is still needed.